# Multi-Agent Actor-Critic for Mixed Cooperative-Competitive Environments

**Ryan Lowe**[*]
McGill University
OpenAI

**Yi Wu**[*]
UC Berkeley

**Aviv Tamar**
UC Berkeley

**Jean Harb**
McGill University
OpenAI

**Pieter Abbeel**
UC Berkeley
OpenAI

**Igor Mordatch**
OpenAI

## Abstract

We explore deep reinforcement learning methods for multi-agent domains. We begin by analyzing the difficulty of traditional algorithms in the multi-agent case: Q-learning is challenged by an inherent non-stationarity of the environment, while policy gradient suffers from a variance that increases as the number of agents grows. We then present an adaptation of actor-critic methods that considers action policies of other agents and is able to successfully learn policies that require complex multi-agent coordination. Additionally, we introduce a training regimen utilizing an ensemble of policies for each agent that leads to more robust multi-agent policies. We show the strength of our approach compared to existing methods in cooperative as well as competitive scenarios, where agent populations are able to discover various physical and informational coordination strategies.

## 1 Introduction

Reinforcement learning (RL) has recently been applied to solve challenging problems, from game playing [23, 28] to robotics [18]. In industrial applications, RL is seeing use in large scale systems such as data center cooling [1]. Most of the successes of RL have been in single agent domains, where modelling or predicting the behaviour of other actors in the environment is largely unnecessary.

However, there are a number of important applications that involve interaction between multiple agents, where emergent behavior and complexity arise from agents co-evolving together. For example, multi-robot control [20], the discovery of communication and language [29, 8, 24], multiplayer games [27], and the analysis of social dilemmas [17] all operate in a multi-agent domain. Related problems, such as variants of hierarchical reinforcement learning [6] can also be seen as a multi-agent system, with multiple levels of hierarchy being equivalent to multiple agents. Additionally, multi-agent self-play has recently been shown to be a useful training paradigm [28, 30]. Successfully scaling RL to environments with multiple agents is crucial to building artificially intelligent systems that can productively interact with humans and each other.

Unfortunately, traditional reinforcement learning approaches such as Q-Learning or policy gradient are poorly suited to multi-agent environments. One issue is that each agent's policy is changing as training progresses, and the environment becomes non-stationary from the perspective of any individual agent (in a way that is not explainable by changes in the agent's own policy). This presents learning stability challenges and prevents the straightforward use of past experience replay, which is

---

[*]Equal contribution. Corresponding authors: `ryan.lowe@cs.mcgill.ca`, `jxwuyi@gmail.com`, `mordatch@openai.com`.

crucial for stabilizing deep Q-learning. Policy gradient methods, on the other hand, usually exhibit very high variance when coordination of multiple agents is required. Alternatively, one can use model-based policy optimization which can learn optimal policies via back-propagation, but this requires a (differentiable) model of the world dynamics and assumptions about the interactions between agents. Applying these methods to competitive environments is also challenging from an optimization perspective, as evidenced by the notorious instability of adversarial training methods [11].

In this work, we propose a general-purpose multi-agent learning algorithm that: (1) leads to learned policies that only use local information (i.e. their own observations) at execution time, (2) does not assume a differentiable model of the environment dynamics or any particular structure on the communication method between agents, and (3) is applicable not only to cooperative interaction but to competitive or mixed interaction involving both physical and communicative behavior. The ability to act in mixed cooperative-competitive environments may be critical for intelligent agents; while competitive training provides a natural curriculum for learning [30], agents must also exhibit cooperative behavior (e.g. with humans) at execution time.

We adopt the framework of centralized training with decentralized execution, allowing the policies to use extra information to ease training, so long as this information is not used at test time. It is unnatural to do this with Q-learning without making additional assumptions about the structure of the environment, as the Q function generally cannot contain different information at training and test time. Thus, we propose a simple extension of actor-critic policy gradient methods where the critic is augmented with extra information about the policies of other agents, while the actor only has access to local information. After training is completed, only the local actors are used at execution phase, acting in a decentralized manner and equally applicable in cooperative and competitive settings. This is a natural setting for multi-agent language learning, as full centralization would not require the development of discrete communication protocols.

Since the centralized critic function explicitly uses the decision-making policies of other agents, we additionally show that agents can learn approximate models of other agents online and effectively use them in their own policy learning procedure. We also introduce a method to improve the stability of multi-agent policies by training agents with an ensemble of policies, thus requiring robust interaction with a variety of collaborator and competitor policies. We empirically show the success of our approach compared to existing methods in cooperative as well as competitive scenarios, where agent populations are able to discover complex physical and communicative coordination strategies.

## 2    Related Work

The simplest approach to learning in multi-agent settings is to use independently learning agents. This was attempted with Q-learning in [34], but does not perform well in practice [22]. As we will show, independently-learning policy gradient methods also perform poorly. One issue is that each agent's policy changes during training, resulting in a non-stationary environment and preventing the naïve application of experience replay. Previous work has attempted to address this by inputting other agent's policy parameters to the Q function [35], explicitly adding the iteration index to the replay buffer, or using importance sampling [9]. Deep Q-learning approaches have previously been investigated in [33] to train competing Pong agents.

The nature of interaction between agents can either be cooperative, competitive, or both and many algorithms are designed only for a particular nature of interaction. Most studied are cooperative settings, with strategies such as optimistic and hysteretic Q function updates [15, 21, 25], which assume that the actions of other agents are made to improve collective reward. Another approach is to indirectly arrive at cooperation via sharing of policy parameters [12], but this requires homogeneous agent capabilities. These algorithms are generally not applicable in competitive or mixed settings. See [26, 4] for surveys of multi-agent learning approaches and applications.

Concurrently to our work, [7] proposed a similar idea of using policy gradient methods with a centralized critic, and test their approach on a StarCraft micromanagement task. Their approach differs from ours in the following ways: (1) they learn a single centralized critic for all agents, whereas we learn a centralized critic for each agent, allowing for agents with differing reward functions including competitive scenarios, (2) we consider environments with explicit communication between agents, (3) they combine recurrent policies with feed-forward critics, whereas our experiments

use feed-forward policies (although our methods are applicable to recurrent policies), (4) we learn continuous policies whereas they learn discrete policies.

Recent work has focused on learning grounded cooperative communication protocols between agents to solve various tasks [29, 8, 24]. However, these methods are usually only applicable when the communication between agents is carried out over a dedicated, differentiable communication channel.

Our method requires explicitly modeling decision-making process of other agents. The importance of such modeling has been recognized by both reinforcement learning [3, 5] and cognitive science communities [10]. [13] stressed the importance of being robust to the decision making process of other agents, as do others by building Bayesian models of decision making. We incorporate such robustness considerations by requiring that agents interact successfully with an ensemble of any possible policies of other agents, improving training stability and robustness of agents after training.

## 3 Background

**Markov Games**   In this work, we consider a multi-agent extension of Markov decision processes (MDPs) called partially observable Markov games [19]. A Markov game for $N$ agents is defined by a set of states $\mathcal{S}$ describing the possible configurations of all agents, a set of actions $\mathcal{A}_1, ..., \mathcal{A}_N$ and a set of observations $\mathcal{O}_1, ..., \mathcal{O}_N$ for each agent. To choose actions, each agent $i$ uses a stochastic policy $\boldsymbol{\pi}_{\theta_i} : \mathcal{O}_i \times \mathcal{A}_i \mapsto [0, 1]$, which produces the next state according to the state transition function $\mathcal{T} : \mathcal{S} \times \mathcal{A}_1 \times ... \times \mathcal{A}_N \mapsto \mathcal{S}$.[2] Each agent $i$ obtains rewards as a function of the state and agent's action $r_i : \mathcal{S} \times \mathcal{A}_i \mapsto \mathbb{R}$, and receives a private observation correlated with the state $\mathbf{o}_i : \mathcal{S} \mapsto \mathcal{O}_i$. The initial states are determined by a distribution $\rho : \mathcal{S} \mapsto [0, 1]$. Each agent $i$ aims to maximize its own total expected return $R_i = \sum_{t=0}^{T} \gamma^t r_i^t$ where $\gamma$ is a discount factor and $T$ is the time horizon.

**Q-Learning and Deep Q-Networks (DQN).**   Q-Learning and DQN [23] are popular methods in reinforcement learning and have been previously applied to multi-agent settings [8, 35]. Q-Learning makes use of an action-value function for policy $\boldsymbol{\pi}$ as $Q^{\boldsymbol{\pi}}(s, a) = \mathbb{E}[R|s^t = s, a^t = a]$. This Q function can be recursively rewritten as $Q^{\boldsymbol{\pi}}(s, a) = \mathbb{E}_{s'}[r(s, a) + \gamma \mathbb{E}_{a' \sim \boldsymbol{\pi}}[Q^{\boldsymbol{\pi}}(s', a')]]$. DQN learns the action-value function $Q^*$ corresponding to the optimal policy by minimizing the loss:

$$\mathcal{L}(\theta) = \mathbb{E}_{s,a,r,s'}[(Q^*(s, a|\theta) - y)^2], \qquad \text{where} \qquad y = r + \gamma \max_{a'} \bar{Q}^*(s', a'), \qquad (1)$$

where $\bar{Q}$ is a target Q function whose parameters are periodically updated with the most recent $\theta$, which helps stabilize learning. Another crucial component of stabilizing DQN is the use of an experience replay buffer $\mathcal{D}$ containing tuples $(s, a, r, s')$.

Q-Learning can be directly applied to multi-agent settings by having each agent $i$ learn an independently optimal function $Q_i$ [34]. However, because agents are independently updating their policies as learning progresses, the environment appears non-stationary from the view of any one agent, violating Markov assumptions required for convergence of Q-learning. Another difficulty observed in [9] is that the experience replay buffer cannot be used in such a setting since in general, $P(s'|s, a, \boldsymbol{\pi}_1, ..., \boldsymbol{\pi}_N) \neq P(s'|s, a, \boldsymbol{\pi}_1', ..., \boldsymbol{\pi}_N')$ when any $\boldsymbol{\pi}_i \neq \boldsymbol{\pi}_i'$.

**Policy Gradient (PG) Algorithms.**   Policy gradient methods are another popular choice for a variety of RL tasks. The main idea is to directly adjust the parameters $\theta$ of the policy in order to maximize the objective $J(\theta) = \mathbb{E}_{s \sim p^{\boldsymbol{\pi}}, a \sim \boldsymbol{\pi}_\theta}[R]$ by taking steps in the direction of $\nabla_\theta J(\theta)$. Using the Q function defined previously, the gradient of the policy can be written as [32]:

$$\nabla_\theta J(\theta) = \mathbb{E}_{s \sim p^{\boldsymbol{\pi}}, a \sim \boldsymbol{\pi}_\theta}[\nabla_\theta \log \boldsymbol{\pi}_\theta(a|s) Q^{\boldsymbol{\pi}}(s, a)], \qquad (2)$$

where $p^{\boldsymbol{\pi}}$ is the state distribution. The policy gradient theorem has given rise to several practical algorithms, which often differ in how they estimate $Q^{\boldsymbol{\pi}}$. For example, one can simply use a sample return $R^t = \sum_{i=t}^{T} \gamma^{i-t} r_i$, which leads to the REINFORCE algorithm [37]. Alternatively, one could learn an approximation of the true action-value function $Q^{\boldsymbol{\pi}}(s, a)$ by e.g. temporal-difference learning [31]; this $Q^{\boldsymbol{\pi}}(s, a)$ is called the *critic* and leads to a variety of *actor-critic* algorithms [31].

Policy gradient methods are known to exhibit high variance gradient estimates. This is exacerbated in multi-agent settings; since an agent's reward usually depends on the actions of many agents,

the reward conditioned only on the agent's own actions (when the actions of other agents are not considered in the agent's optimization process) exhibits much more variability, thereby increasing the variance of its gradients. Below, we show a simple setting where the probability of taking a gradient step in the correct direction decreases exponentially with the number of agents.

**Proposition 1.** *Consider $N$ agents with binary actions: $P(a_i = 1) = \theta_i$, where $R(a_1, \ldots, a_N) = \mathbf{1}_{a_1 = \cdots = a_N}$. We assume an uninformed scenario, in which agents are initialized to $\theta_i = 0.5 \ \forall i$. Then, if we are estimating the gradient of the cost $J$ with policy gradient, we have:*

$$P(\langle \hat{\nabla} J, \nabla J \rangle > 0) \propto (0.5)^N$$

*where $\hat{\nabla} J$ is the policy gradient estimator from a single sample, and $\nabla J$ is the true gradient.*

*Proof.* See Appendix. $\square$

The use of baselines, such as value function baselines typically used to ameliorate high variance, is problematic in multi-agent settings due to the non-stationarity issues mentioned previously.

**Deterministic Policy Gradient (DPG) Algorithms.** It is also possible to extend the policy gradient framework to deterministic policies $\boldsymbol{\mu}_\theta : \mathcal{S} \mapsto \mathcal{A}$. In particular, under certain conditions we can write the gradient of the objective $J(\theta) = \mathbb{E}_{s \sim p^\mu}[R(s, a)]$ as:

$$\nabla_\theta J(\theta) = \mathbb{E}_{s \sim \mathcal{D}}[\nabla_\theta \boldsymbol{\mu}_\theta(a|s) \nabla_a Q^{\boldsymbol{\mu}}(s, a)|_{a = \boldsymbol{\mu}_\theta(s)}] \quad (3)$$

Since this theorem relies on $\nabla_a Q^{\boldsymbol{\mu}}(s, a)$, it requires that the action space $\mathcal{A}$ (and thus the policy $\boldsymbol{\mu}$) be continuous.

*Deep deterministic policy gradient* (DDPG) is a variant of DPG where the policy $\boldsymbol{\mu}$ and critic $Q^{\boldsymbol{\mu}}$ are approximated with deep neural networks. DDPG is an off-policy algorithm, and samples trajectories from a replay buffer of experiences that are stored throughout training. DDPG also makes use of a target network, as in DQN [23].

# 4   Methods

## 4.1   Multi-Agent Actor Critic

We have argued in the previous section that naïve policy gradient methods perform poorly in simple multi-agent settings, and this is supported in our experiments in Section 5. Our goal in this section is to derive an algorithm that works well in such settings. However, we would like to operate under the following constraints: (1) the learned policies can only use local information (i.e. their own observations) at execution time, (2) we do not assume a differentiable model of the environment dynamics, unlike in [24], and (3) we do not assume any particular structure on the communication method between agents (that is, we don't assume a differentiable communication channel). Fulfilling the above desiderata would provide a general-purpose multi-agent learning algorithm that could be applied not just to cooperative games with explicit communication channels, but competitive games and games involving only physical interactions between agents.

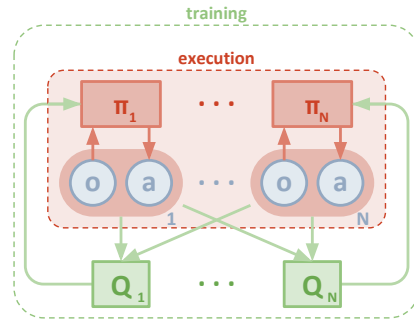

Figure 1: Overview of our multi-agent decentralized actor, centralized critic approach.

Similarly to [8], we accomplish our goal by adopting the framework of centralized training with decentralized execution. Thus, we allow the policies to use extra information to ease training, so long as this information is not used at test time. It is unnatural to do this with Q-learning, as the Q function generally cannot contain different information at training and test time. Thus, we propose a simple extension of actor-critic policy gradient methods where the critic is augmented with extra information about the policies of other agents.

More concretely, consider a game with $N$ agents with policies parameterized by $\boldsymbol{\theta} = \{\theta_1, ..., \theta_N\}$, and let $\boldsymbol{\pi} = \{\boldsymbol{\pi}_1, ..., \boldsymbol{\pi}_N\}$ be the set of all agent policies. Then we can write the gradient of the

expected return for agent $i$, $J(\theta_i) = \mathbb{E}[R_i]$ as:

$$\nabla_{\theta_i} J(\theta_i) = \mathbb{E}_{s \sim p^{\mu}, a_i \sim \boldsymbol{\pi}_i}[\nabla_{\theta_i} \log \boldsymbol{\pi}_i(a_i|o_i) Q_i^{\boldsymbol{\pi}}(\mathbf{x}, a_1, ..., a_N)]. \tag{4}$$

Here $Q_i^{\boldsymbol{\pi}}(\mathbf{x}, a_1, ..., a_N)$ is a *centralized action-value function* that takes as input the actions of all agents, $a_1, \ldots, a_N$, in addition to some state information $\mathbf{x}$, and outputs the Q-value for agent $i$. In the simplest case, $\mathbf{x}$ could consist of the observations of all agents, $\mathbf{x} = (o_1, ..., o_N)$, however we could also include additional state information if available. Since each $Q_i^{\boldsymbol{\pi}}$ is learned separately, agents can have arbitrary reward structures, including conflicting rewards in a competitive setting.

We can extend the above idea to work with deterministic policies. If we now consider $N$ continuous policies $\boldsymbol{\mu}_{\theta_i}$ w.r.t. parameters $\theta_i$ (abbreviated as $\boldsymbol{\mu}_i$), the gradient can be written as:

$$\nabla_{\theta_i} J(\boldsymbol{\mu}_i) = \mathbb{E}_{\mathbf{x}, a \sim \mathcal{D}}[\nabla_{\theta_i} \boldsymbol{\mu}_i(a_i|o_i) \nabla_{a_i} Q_i^{\boldsymbol{\mu}}(\mathbf{x}, a_1, ..., a_N)|_{a_i = \boldsymbol{\mu}_i(o_i)}], \tag{5}$$

Here the experience replay buffer $\mathcal{D}$ contains the tuples $(\mathbf{x}, \mathbf{x}', a_1, \ldots, a_N, r_1, \ldots, r_N)$, recording experiences of all agents. The centralized action-value function $Q_i^{\boldsymbol{\mu}}$ is updated as:

$$\mathcal{L}(\theta_i) = \mathbb{E}_{\mathbf{x}, a, r, \mathbf{x}'}[(Q_i^{\boldsymbol{\mu}}(\mathbf{x}, a_1, \ldots, a_N) - y)^2], \quad y = r_i + \gamma Q_i^{\boldsymbol{\mu}'}(\mathbf{x}', a_1', \ldots, a_N')\big|_{a_j' = \boldsymbol{\mu}_j'(o_j)}, \tag{6}$$

where $\boldsymbol{\mu}' = \{\boldsymbol{\mu}_{\theta_1'}, ..., \boldsymbol{\mu}_{\theta_N'}\}$ is the set of target policies with delayed parameters $\theta_i'$. As shown in Section 5, we find the centralized critic with deterministic policies works very well in practice, and refer to it as *multi-agent deep deterministic policy gradient* (MADDPG). We provide the description of the full algorithm in the Appendix.

A primary motivation behind MADDPG is that, if we know the actions taken by all agents, the environment is stationary even as the policies change, since $P(s'|s, a_1, ..., a_N, \boldsymbol{\pi}_1, ..., \boldsymbol{\pi}_N) = P(s'|s, a_1, ..., a_N) = P(s'|s, a_1, ..., a_N, \boldsymbol{\pi}_1', ..., \boldsymbol{\pi}_N')$ for any $\boldsymbol{\pi}_i \neq \boldsymbol{\pi}_i'$. This is not the case if we do not explicitly condition on the actions of other agents, as done for most traditional RL methods.

Note that we require the policies of other agents to apply an update in Eq. 6. Knowing the observations and policies of other agents is not a particularly restrictive assumption; if our goal is to train agents to exhibit complex communicative behaviour in simulation, this information is often available to all agents. However, we can relax this assumption if necessary by learning the policies of other agents from observations — we describe a method of doing this in Section 4.2.

## 4.2 Inferring Policies of Other Agents

To remove the assumption of knowing other agents' policies, as required in Eq. 6, each agent $i$ can additionally maintain an approximation $\hat{\boldsymbol{\mu}}_{\phi_i^j}$ (where $\phi$ are the parameters of the approximation; henceforth $\hat{\boldsymbol{\mu}}_i^j$) to the true policy of agent $j$, $\boldsymbol{\mu}_j$. This approximate policy is learned by maximizing the log probability of agent $j$'s actions, with an entropy regularizer:

$$\mathcal{L}(\phi_i^j) = -\mathbb{E}_{o_j, a_j}\left[\log \hat{\boldsymbol{\mu}}_i^j(a_j|o_j) + \lambda H(\hat{\boldsymbol{\mu}}_i^j)\right], \tag{7}$$

where $H$ is the entropy of the policy distribution. With the approximate policies, $y$ in Eq. 6 can be replaced by an approximate value $\hat{y}$ calculated as follows:

$$\hat{y} = r_i + \gamma Q_i^{\boldsymbol{\mu}'}(\mathbf{x}', \hat{\boldsymbol{\mu}}_i'^1(o_1), \ldots, \boldsymbol{\mu}_i'(o_i), \ldots, \hat{\boldsymbol{\mu}}_i'^N(o_N)), \tag{8}$$

where $\hat{\boldsymbol{\mu}}_i'^j$ denotes the target network for the approximate policy $\hat{\boldsymbol{\mu}}_i^j$. Note that Eq. 7 can be optimized in a completely online fashion: before updating $Q_i^{\boldsymbol{\mu}}$, the centralized Q function, we take the latest samples of each agent $j$ from the replay buffer to perform a single gradient step to update $\phi_i^j$. We also input the action log probabilities of each agent directly into $Q$, rather than sampling.

## 4.3 Agents with Policy Ensembles

A recurring problem in multi-agent reinforcement learning is the environment non-stationarity due to the agents' changing policies. This is particularly true in competitive settings, where agents can derive a strong policy by overfitting to the behavior of their competitors. Such policies are undesirable as they are brittle and may fail when the competitors alter their strategies.

To obtain multi-agent policies that are more robust to changes in the policy of competing agents, we propose to train a collection of $K$ different sub-policies. At each episode, we randomly select one particular sub-policy for each agent to execute. Suppose that policy $\boldsymbol{\mu}_i$ is an ensemble of $K$ different sub-policies with sub-policy $k$ denoted by $\boldsymbol{\mu}_{\theta_i^{(k)}}$ (denoted as $\boldsymbol{\mu}_i^{(k)}$). For agent $i$, we are then maximizing the ensemble objective: $J_e(\boldsymbol{\mu}_i) = \mathbb{E}_{k\sim\text{unif}(1,K),s\sim p^{\boldsymbol{\mu}},a\sim\boldsymbol{\mu}_i^{(k)}}\left[R_i(s,a)\right].$

Since different sub-policies will be executed in different episodes, we maintain a replay buffer $\mathcal{D}_i^{(k)}$ for each sub-policy $\boldsymbol{\mu}_i^{(k)}$ of agent $i$. Accordingly, we can derive the gradient of the ensemble objective with respect to $\theta_i^{(k)}$ as follows:

$$\nabla_{\theta_i^{(k)}} J_e(\boldsymbol{\mu}_i) = \frac{1}{K} \mathbb{E}_{\mathbf{x},a\sim\mathcal{D}_i^{(k)}} \left[ \nabla_{\theta_i^{(k)}} \boldsymbol{\mu}_i^{(k)}(a_i|o_i) \nabla_{a_i} Q^{\boldsymbol{\mu}_i}\left(\mathbf{x}, a_1, \ldots, a_N\right)\Big|_{a_i=\boldsymbol{\mu}_i^{(k)}(o_i)} \right]. \qquad (9)$$

## 5 Experiments[2]

### 5.1 Environments

To perform our experiments, we adopt the grounded communication environment proposed in [24], which consists of $N$ agents and $L$ landmarks inhabiting a two-dimensional world with continuous space and discrete time[2]. Agents may take physical actions in the environment and communication actions that get broadcasted to other agents. Unlike [24], we do not assume that all agents have identical action and observation spaces, or act according to the same policy $\boldsymbol{\pi}$. We also consider games that are both cooperative (all agents must maximize a shared return) and competitive (agents have conflicting goals). Some environments require explicit communication between agents in order to achieve the best reward, while in other environments agents can only perform physical actions. We provide details for each environment below.

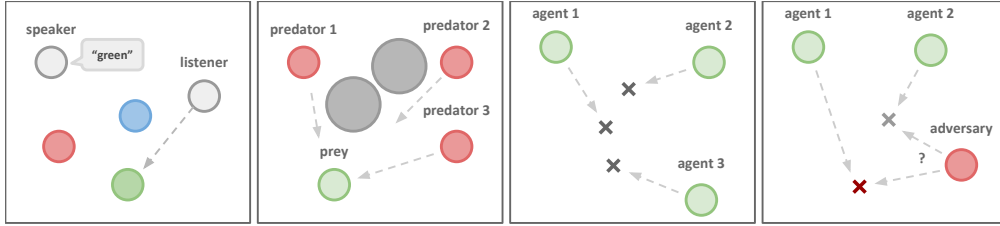

Figure 2: Illustrations of the experimental environment and some tasks we consider, including a) *Cooperative Communication* b) *Predator-Prey* c) *Cooperative Navigation* d) *Physical Deception*. See webpage for videos of all experimental results.

**Cooperative communication.** This task consists of two cooperative agents, a speaker and a listener, who are placed in an environment with three landmarks of differing colors. At each episode, the listener must navigate to a landmark of a particular color, and obtains reward based on its distance to the correct landmark. However, while the listener can observe the relative position and color of the landmarks, it does not know which landmark it must navigate to. Conversely, the speaker's observation consists of the correct landmark color, and it can produce a communication output at each time step which is observed by the listener. Thus, the speaker must learn to output the landmark colour based on the motions of the listener. Although this problem is relatively simple, as we show in Section 5.2 it poses a significant challenge to traditional RL algorithms.

**Cooperative navigation.** In this environment, agents must cooperate through physical actions to reach a set of $L$ landmarks. Agents observe the relative positions of other agents and landmarks, and are collectively rewarded based on the proximity of any agent to each landmark. In other words, the agents have to 'cover' all of the landmarks. Further, the agents occupy significant physical space and are penalized when colliding with each other. Our agents learn to infer the landmark they must cover, and move there while avoiding other agents.

[2] Videos of our experimental results can be viewed at https://sites.google.com/site/multiagentac/
[2] The environments are publicly available: https://github.com/openai/multiagent-particle-envs

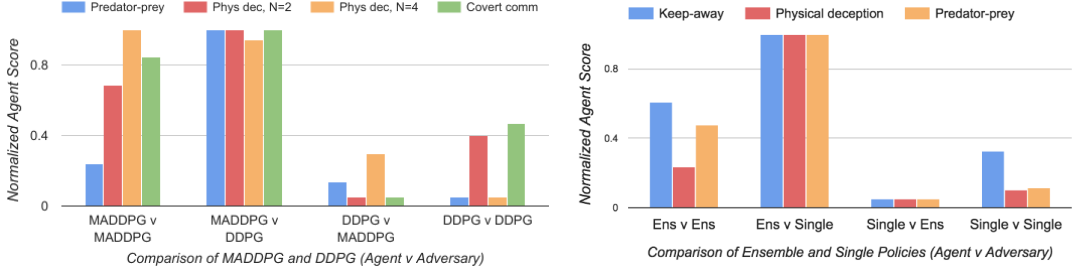

Figure 3: Comparison between MADDPG and DDPG (left), and between single policy MADDPG and ensemble MADDPG (right) on the competitive environments. Each bar cluster shows the 0-1 normalized score for a set of competing policies (agent v adversary), where a higher score is better for the agent. In all cases, MADDPG outperforms DDPG when directly pitted against it, and similarly for the ensemble against the single MADDPG policies. Full results are given in the Appendix.

**Keep-away.** This scenario consists of $L$ landmarks including a target landmark, $N$ cooperating agents who know the target landmark and are rewarded based on their distance to the target, and $M$ *adversarial* agents who must prevent the cooperating agents from reaching the target. Adversaries accomplish this by physically pushing the agents away from the landmark, temporarily occupying it. While the adversaries are also rewarded based on their distance to the target landmark, they do not know the correct target; this must be inferred from the movements of the agents.

**Physical deception.** Here, $N$ agents cooperate to reach a single target landmark from a total of $N$ landmarks. They are rewarded based on the minimum distance of any agent to the target (so only one agent needs to reach the target landmark). However, a lone adversary also desires to reach the target landmark; the catch is that the adversary does not know which of the landmarks is the correct one. Thus the cooperating agents, who are penalized based on the adversary distance to the target, learn to spread out and cover all landmarks so as to deceive the adversary.

**Predator-prey.** In this variant of the classic predator-prey game, $N$ slower cooperating agents must chase the faster adversary around a randomly generated environment with $L$ large landmarks impeding the way. Each time the cooperative agents collide with an adversary, the agents are rewarded while the adversary is penalized. Agents observe the relative positions and velocities of the agents, and the positions of the landmarks.

**Covert communication.** This is an adversarial communication environment, where a speaker agent ('Alice') must communicate a message to a listener agent ('Bob'), who must reconstruct the message at the other end. However, an adversarial agent ('Eve') is also observing the channel, and wants to reconstruct the message — Alice and Bob are penalized based on Eve's reconstruction, and thus Alice must encode her message using a randomly generated *key*, known only to Alice and Bob. This is similar to the cryptography environment considered in [2].

## 5.2   Comparison to Decentralized Reinforcement Learning Methods

We implement MADDPG and evaluate it on the environments presented in Section 5.1. Unless otherwise specified, our policies are parameterized by a two-layer ReLU MLP with 64 units per layer. To support discrete communication messages, we use the Gumbel-Softmax estimator [14]. To evaluate the quality of policies learned in competitive settings, we pitch MADDPG agents against DDPG agents, and compare the resulting success of the agents and adversaries in the environment. We train our models until convergence, and then evaluate them by averaging various metrics for 1000 further iterations. We provide the tables and details of our results on all environments in the Appendix, and summarize them here.

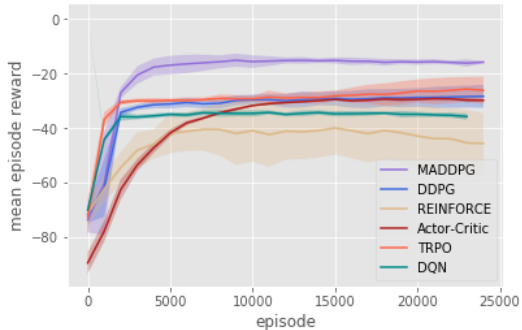

Figure 4: The reward of MADDPG against traditional RL approaches on cooperative communication after 25000 episodes.

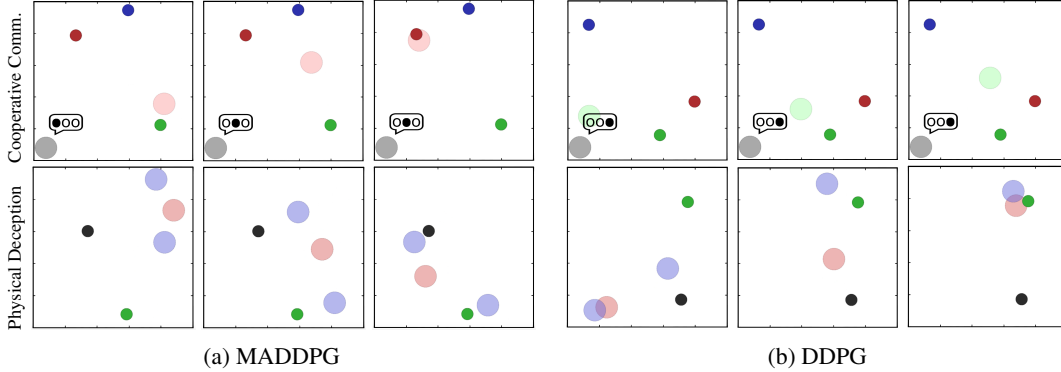

(a) MADDPG                         (b) DDPG

Figure 5: Comparison between MADDPG (left) and DDPG (right) on the cooperative communication (CC) and physical deception (PD) environments at $t = 0$, $5$, and $25$. Small dark circles indicate landmarks. In CC, the grey agent is the speaker, and the color of the listener indicates the target landmark. In PD, the blue agents are trying to deceive the red adversary, while covering the target landmark (in green). MADDPG learns the correct behavior in both cases: in CC the speaker learns to output the target landmark color to direct the listener, while in PD the agents learn to cover both landmarks to confuse the adversary. DDPG (and other RL algorithms) struggles in these settings: in CC the speaker always repeats the same utterance and the listener moves to the middle of the landmarks, and in PP one agent greedily pursues the green landmark (and is followed by the adversary) while the othe agent scatters. See video for full trajectories.

We first examine the cooperative communication scenario. Despite the simplicity of the task (the speaker only needs to learn to output its observation), traditional RL methods such as DQN, Actor-Critic, a first-order implementation of TRPO, and DDPG all fail to learn the correct behaviour (measured by whether the listener is within a short distance from the target landmark). In practice we observed that the listener learns to ignore the speaker and simply moves to the middle of all observed landmarks. We plot the learning curves over 25000 episodes for various approaches in Figure 4.

We hypothesize that a primary reason for the failure of traditional RL methods in this (and other) multi-agent settings is the lack of a consistent gradient signal. For example, if the speaker utters the correct symbol while the listener moves in the wrong direction, the speaker is penalized. This problem is exacerbated as the number of time steps grows: we observed that traditional policy gradient methods can learn when the objective of the listener is simply to reconstruct the observation of the speaker in a single time step, or if the initial positions of agents and landmarks are fixed and evenly distributed. This indicates that many of the multi-agent methods previously proposed for scenarios with short time horizons (e.g. [16]) may not generalize to more complex tasks.

Conversely, MADDPG agents can learn coordinated behaviour more easily via the centralized critic. In the cooperative communication environment, MADDPG is able to reliably learn the correct listener and speaker policies, and the listener is often (84.0% of the time) able to navigate to the target.

A similar situation arises for the physical deception task: when the cooperating agents are trained with MADDPG, they are able to successfully deceive the adversary by covering all of the landmarks around 94% of the time when $L = 2$ (Figure 5). Furthermore, the adversary success is quite low, especially when the adversary is trained with DDPG (16.4% when $L = 2$). This contrasts sharply with the behaviour learned by the cooperating DDPG agents, who are unable to deceive MADDPG adversaries in any scenario, and do not even deceive other DDPG agents when $L = 4$.

While the cooperative navigation and predator-prey tasks have a less stark divide between success and failure, in both cases the MADDPG agents outperform the DDPG agents. In cooperative navigation, MADDPG agents have a slightly smaller average distance to each landmark, but have almost half the average number of collisions per episode (when $N = 2$) compared to DDPG agents due to the ease of coordination. Similarly, MADDPG predators are far more successful at chasing DDPG prey (16.1 collisions/episode) than the converse (10.3 collisions/episode).

In the covert communication environment, we found that Bob trained with both MADDPG and DDPG out-performs Eve in terms of reconstructing Alice's message. However, Bob trained with MADDPG achieves a larger relative success rate compared with DDPG (52.4% to 25.1%). Further, only Alice trained with MADDPG can encode her message such that Eve achieves near-random

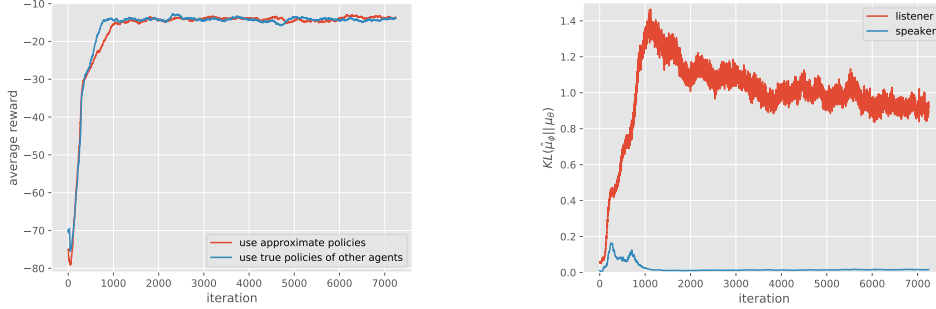

Figure 6: Effectiveness of learning by approximating policies of other agents in the cooperative communication scenario. *Left:* plot of the reward over number of iterations; MADDPG agents quickly learn to solve the task when approximating the policies of others. *Right:* KL divergence between the approximate policies and the true policies.

reconstruction accuracy. The learning curve (a sample plot is shown in Appendix) shows that the oscillation due to the competitive nature of the environment often cannot be overcome with common decentralized RL methods. We emphasize that we do not use any of the tricks required for the cryptography environment from [2], including modifying Eve's loss function, alternating agent and adversary training, and using a hybrid 'mix & transform' feed-forward and convolutional architecture.

### 5.3 Effect of Learning Polices of Other Agents

We evaluate the effectiveness of learning the policies of other agents in the cooperative communication environment, following the same hyperparameters as the previous experiments and setting $\lambda = 0.001$ in Eq. 7. The results are shown in Figure 6. We observe that despite not fitting the policies of other agents perfectly (in particular, the approximate listener policy learned by the speaker has a fairly large KL divergence to the true policy), learning with approximated policies is able to achieve the same success rate as using the true policy, without a significant slowdown in convergence.

### 5.4 Effect of Training with Policy Ensembles

We focus on the effectiveness of policy ensembles in competitive environments, including keep-away, cooperative navigation, and predator-prey. We choose $K = 3$ sub-policies for the keep-away and cooperative navigation environments, and $K = 2$ for predator-prey. To improve convergence speed, we enforce that the cooperative agents should have the same policies at each episode, and similarly for the adversaries. To evaluate the approach, we measure the performance of ensemble policies and single policies in the roles of both agent and adversary. The results are shown on the right side of Figure 3. We observe that agents with policy ensembles are stronger than those with a single policy. In particular, when pitting ensemble agents against single policy adversaries (second to left bar cluster), the ensemble agents outperform the adversaries by a large margin compared to when the roles are reversed (third to left bar cluster).

## 6 Conclusions and Future Work

We have proposed a multi-agent policy gradient algorithm where agents learn a centralized critic based on the observations and actions of all agents. Empirically, our method outperforms traditional RL algorithms on a variety of cooperative and competitive multi-agent environments. We can further improve the performance of our method by training agents with an ensemble of policies, an approach we believe to be generally applicable to any multi-agent algorithm.

One downside to our approach is that the input space of $Q$ grows linearly (depending on what information is contained in $\mathbf{x}$) with the number of agents $N$. This could be remedied in practice by, for example, having a modular Q function that only considers agents in a certain neighborhood of a given agent. We leave this investigation to future work.

**Acknowledgements**

The authors would like to thank Jacob Andreas, Smitha Milli, Jack Clark, Jakob Foerster, and others at OpenAI and UC Berkeley for interesting discussions related to this paper, as well as Jakub Pachocki, Yura Burda, and Joelle Pineau for comments on the paper draft. We thank Tambet Matiisen for providing the code base that was used for some early experiments associated with this paper. Ryan Lowe is supported in part by a Vanier CGS Scholarship and the Samsung Advanced Institute of Technology. Finally, we'd like to thank OpenAI for fostering an engaging and productive research environment.

## Footnotes

[2]To minimize notation we will often omit $\theta$ from the subscript of $\boldsymbol{\pi}$.

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
