[Reviews · NeurIPS 2017]

Reviewer 1



This paper presents an actor-critic approach for multi-agent reinforcement learning in both cooperative and competitive (adversarial) scenarios. The contribution is the multi-agent deterministic policy gradient algorithm, with a different policy per agent, and with a different centralized (accessing actions from all agents) critic per agent. They ran experiments on 6 cooperative and/or competitive small environments. The paper is overall well written, and present a significant contribution, but too many of the claim-supporting experiments are in the Appendix. My biggest concern is with the experimental details of the A2C/DQN/DDPG implementations. In particular, it seems to me like there should be variants of these trained in multi-agent and in single-agent-multiple-effectors, to control for what the multi-agent setting is helping with (or making more difficult) in the reward attribution (or structure) inference. I believe that this is good enough for publication at NIPS, even though I would appreciate a finer grained analysis of which components (ablation) from MADDPG are necessary to solve these simple tasks, and/or why the other methods perform worse.

Reviewer 2



Summary ----------------- The paper presents a novel actor-critic algorithm, named MADDPG, for both cooperative and competitive multiagent problems. MADDPG relies on a number of key ideas: 1) The action value functions are learned in a 'centralized' manner, meaning that it takes into account the actions of all other players. This allows to evaluate the effect of the joint policy on each agents long term reward. To remove the need of knowing other agents' actions, the authors suggest that each agent could learn an approximate model of their policies. 2) The learned policies can only use local information, so that they can be used by each agent without relying on further communication. 3) It learns a collection of policies for each agent, rather than just one. At each episode during the learning process, each agent draws uniformaly a policy from its ensemble. Sampling a policy from an ensemble reduces the non-statinoarity caused by multiple agents learning at the same time. Several interesting, both cooperative and competitive, are considered to evaluate the proposed algorithm. Simulation results show the benefit of using ensembles and improvement with respect to independent DDPG agents. Major comments ----------------- The main algorithmic contribution of the paper seems to be the policy ensemble to surmount nonstationarity. This is an interesting idea. However, very little insight is provided. Do the authors have any hypothesis of why this seems to work well? The environments considered in Sec. 5 constitute an interesting benchmark. However, the proposed MADDPG algorithm is compared against independent DDPG agents. Although this validates the effectiveness of the algorithm, it would be more interseting to compare with centralized DDPG, as well as with other centralized critic approaches (e.g., [6]), and other ensemble RL algorithms. The organization of the paper is somehow unbalanced. For instance, it reviews well known methods, while there are some important points that remain unclear. In particular, it seems that the algorithm learns as many action value functions as agents, so that it learns a parameter theta_i per agent. However, in lines 333-334, the authors mention that the complexity grows linearly for some Q (without subscript), rather than considering the computational burden of learning a set of action value functions {Q_i}. Minor comments ----------------- - Sentence in lines 47-48 is difficult to understand. - The use of bold font notation seems arbitrary. Why are { pi_i } bold while { theta_i } aren't ? - Word 'Reference' should precede '[11]' in line 81.

Reviewer 3



The paper proposes a novel application an actor-critic algorithm to multi-agent problems with "centralized training and distributed testing" constraint. The trick is that the critic can be trained in centralized way (have access to all observations and policies) because it is not used during testing. Experiments on diverse set of tasks show it outperforms a model trained in distributed way. My concerns: - Centralized training and distributed testing assumption is not very realistic. The paper doesn't mention any real-world application that allows centralized training but prohibits centralized testing. - Exponential growth of the variance of policy gradient with the number of agents is shown, but nothing is said about the variance of the proposed method. In the considered example, any algorithm will require exponential number of samples to observe a successful episode. - Only single environment is used in the experiments, which makes the results less convincing. - Missing details: what is X is fed to Q in the experiments? All observations combined? - Most of the results are not in the paper itself.

Reviewer 4



The paper describes a centralised multiagent training algorithm leading to decentralized individual policies. Overall the paper is nicely written and is extremely easy to understand. The paper identifies two problems faced when training agents i the multi-agent setting: 1) the problem of extreme non-stationarity faced when naive single-agent algorithms are used in the multiagent setting; 2) the ability for policies co-adapt and not to generalize well to unseen oponent policies. In order to address the first problem, the authors suggest using a special actor-critic algorithm which use local actors but a global critic. Overall this is a very simple and elegant suggestion for the problem setting. While the suggested method should indeed reduce variance (even in the single agent case) Eq (4) shows that it will also add a lot of bias. Why was not it possible to use a centralized state value function Vi(x, a1, ... an) instead of Qi(...) and use it as a baseline in the standard policy gradient to train a local actor? As this baseline would be a more correct baseline, the discussed variance problem in the multi-agent case should be reduced. Also, if one is going down the variance reduction path, another plausible option is to simply minimize E_{s} sum_{a1} pi1(x, a1)Q(x, a1, ... an) instead of E_{s,a1} log pi1(x, a1)Q(x, a1, ... an) (note the lack of logarithm and E_{a1} being replaced as sum_{a1}). The dissapearing log term has the potential to further reduce variance, as d log(pi(x)) / dx = pi'(x)/pi(x) which ca be very large for small pi. In order to address the second problem, the authors suggested to use ensamble training of policies. Overall I see this paper as a nice contribution to the field and I would like to see it accepted. However, I would like the authors to discuss what kind of policy evaluation algorithm they used in order to evaluate Qi.